# Recent Advances in Miscanthus Macromolecule Conversion: A Brief Overview

**DOI:** 10.3390/ijms241613001

**Published:** 2023-08-20

**Authors:** Galina F. Mironova, Vera V. Budaeva, Ekaterina A. Skiba, Yulia A. Gismatulina, Ekaterina I. Kashcheyeva, Gennady V. Sakovich

**Affiliations:** Laboratory of Bioconversion, Institute for Problems of Chemical and Energetic Technologies, Siberian Branch of the Russian Academy of Sciences (IPCET SB RAS), 659322 Biysk, Russia; yur_galina@mail.ru (G.F.M.); eas08988@mail.ru (E.A.S.); julja.gismatulina@rambler.ru (Y.A.G.); makarova@ipcet.ru (E.I.K.);

**Keywords:** miscanthus, renewable polymers, biofuel, bacterial cellulose, biopolymers, enzymes, platform molecules

## Abstract

Miscanthus is a valuable renewable feedstock and has a significant potential for the manufacture of diverse biotechnology products based on macromolecules such as cellulose, hemicelluloses and lignin. Herein, we overviewed the state-of-the art of research on the conversion of miscanthus polymers into biotechnology products comprising low-molecular compounds and macromolecules: bioethanol, biogas, bacterial cellulose, enzymes (cellulases, laccases), lactic acid, lipids, fumaric acid and polyhydroxyalkanoates. The present review aims to assess the potential of converting miscanthus polymers in order to develop sustainable technologies.

## 1. Introduction

The perpetually increasing atmospheric carbon dioxide and global warming are a serious threat to humankind. Hence, actions are required to mitigate the climate change consequences, and there is a need for the transition to a low-carbon economy in which biomass is the most common and available source of carbon [1].

A good many researchers consider the issue of global greenhouse gas emissions from the standpoint of trading and policy [2,3,4,5,6,7], which is undoubtedly important to combat the climate change. The other research studies are focused on a quantitative evaluation of the potential of various productions to minimize the consequences or reduce the CO_2_ emissions; for example, the estimations of biomass utilization for transport, power engineering, construction and iron-and-steel industry [8].

Miscanthus is a bio-pump [9] and has the potential of greenhouse gas emission reduction through soil carbon assimilation [10].

The studies [11,12,13] reported valuable results evaluating the life cycle of heat, electric power, ethanol and biogas productions from miscanthus, and demonstrated that the miscanthus cultivation and the manufacture of commodities from miscanthus are a good option for carbon footprint mitigation.

Miscanthus is a perennial rhizomatous grass with a high yield capacity and low nutrient requirements. Miscanthus has a life span up to 20 years, which is an advantage over annual plants. The merits of *Miscanthus × giganteus* may also include the anatomy of its stalks whose bast layer does not contain long fibers unlike some bast plants that require pruning of their bast fibers (for example, flax and hemp) [14]. Compared to other perennial crops, miscanthus yields a higher content of dry matter. Once planted, miscanthus requires no fertilizers or special care in the field but annual harvesting with standard farm machinery [15]. This crop with a high water use efficiency and ability to adapt to severe conditions along with its environmental functions such as soil remediation may have a vital part in the bioeconomic development of any nation [14,15]. Miscanthus is a frost-resistant crop and can grow on marginal, salinized and unused lands [16]. Considering the probability of further depletion of the world forest areas and the limitation on wood procurement due to the environment-conserving role of forests, miscanthus is being more frequently viewed as a potential feedstock to replace some of softwood and hardwood [14].

About 123,000 ha are utilized for the miscanthus biomass production across the world. The largest area is located in China, where approx. 100,000 ha are occupied by *M. lutarioriparius* in the wildlife at the Dongting Lake. The biomass yields constitute about 12 t/ha/year [17].

The machine learning study results [18] showed that globally there exist 3068.25 million ha marginal land resources eligible for *M. × giganteus* cultivation, which are basically located in Africa (902.05 million ha), Asia (620.32 million ha), South America (547.60 million ha) and North America (529.26 million ha). The countries with the largest land resources, Russia and Brazil, hold the first and second places based on the amount of marginal lands suitable for *M. × giganteus*, with areas of 373.35 and 332.37 million ha, respectively.

Miscanthus is a valuable renewable feedstock and has a significant potential for the manufacture of diverse biotechnology products based on macromolecules such as cellulose, hemicelluloses and lignin. The studies on the miscanthus chemical composition compared to the diverse vegetable world are constantly developing and show the advantages of miscanthus over many lignocellulosic resources, particularly by the content of cellulose, a polymer that is the most valuable for conversion.

Hong et al. [19] reported that the cellulose content of *M. sinensis* (42.3–43.7%) comes closer to that of bamboo (41.0–49.1%), softwood (40–52%) and hardwood (38–56%) growing in the same location. The later study by Klímek et al. [20] showed that the cellulose content of miscanthus (36–38%) is slightly lower than that of Spruce wood (43%). As shown by Wójciak et al. [21], *M. × giganteus* stems (44.6%) are richer in cellulose than birch wood (43.7%).

Doczekalska et al. [22] in a study on the chemical composition of three *Miscanthus* species (*M. giganteus*, *M. sacchariflorus* and *M. sinensis*) reported the content of cellulose to range from 44.12% to 45.12%, while in switchgrass (*Panicum virgatum*), it was 40.30%. It was found that miscanthu*s* is superior to millet in both the cellulose content and yield capacity. Waliszewska et al. [23] reliably showed that miscanthus contains either much more cellulose or the same percentage as perennial grasses such as reed canary grass (*Phalaris arundinacea L.*), wood small-reed (*Calamagrostis epigejos L. Roth*), common reed (*Phragmites australis Cav.*), couch grass (*Elymus repens L. Gould.*), downy brome (*Bromus tectorum L*.), false oat-grass (*Arrhenatherum elatius (L.*) *P. Beauv. ex J. Presl & C. Presl*), common bent (*Agrostis capillaris L.*), sweet vernal grass (*Anthoxanthum odoratum L*.), cock’s foot (*Dactylis glomerata L*.) and velvet grass (*Holcus lanatus L*.) whose cellulose contents range from 33.38 to 38.68%.

Xu et al. [24] determined quite a wide content range from 29.79 to 48.52% for cellulose, 15.71 to 34.23% for hemicelluloses and 13.01 to 23.75% for lignin in miscanthus accessions in China, and also reported that miscanthus is closer in lignocellulose to wood materials whose cellulose, hemicellulose and lignin contents range from ~30 to 50, 10 to 40 and 5 to 30%, respectively, as compared to other crops.

The present review aimed to present the state-of-the-art research on the conversion of miscanthus (without division into varieties), highlighting biotechnology products in more detail. For this, the international databases were searched for related literature. The literature search was run chiefly for the last five years, with earlier works cited as well. This is the first review on experimentally validated biotechnology approaches to miscanthus conversion, and it is brief but comprehensive and quotes valuable references for the evaluation of the potential for the conversion of miscanthus polymers and for the development of sustainable technologies. In addition, examples of other feedstock sources for biotechnology products are also outlined herein for a comparison purpose.

## 2. Core Directions in Miscanthus Research

Before we look into a narrower direction within the scope of this review, it is worth mentioning the key miscanthus research trends.

### 2.1. Miscanthus Selection

*M. × giganteus* is the most common worldwide among the *Miscanthus* species. The high yield capacity (10 t/ha/year) and life span (15–20 years) make miscanthus a promising bioenergy crop and an effective tool to combat the climate change. However, *M. × giganteus* is not free of shortcomings, i.e., it is sensitive to cold winter temperatures and drought, can only be reproduced through rhizome division, has a poor genetic diversity and is susceptible to soil pathogens. Hence, the other *Miscanthus* species and cultivars have become valuable sources of the genetic material for intraspecific and interspecific breeding. In breeding, a special focus is placed on achieving a higher yield capacity, quality and tolerance to antibiotic stressors. For instance, despite having a poorer aboveground biomass yield compared to *M. × giganteus*, *M. sinensis* is more tolerant to water stress and, hence, is more suitable for cultivation in a drier climate. *M. lutarioriparius* offers a high yield of biomass but is less resistant to cold and drought, and is therefore more suitable for regions that are less exposed to frequent water deficiency [25].

Since the chemical composition of the feedstock is essential for the miscanthus conversion, Table 1 outlines exactly this aspect for some *Miscanthus* species from different geographical locations, as reported in the recent studies. 

Because of miscanthus having a rich genetic diversity, its lignocellulosic content varies widely; yet, many *Miscanthus* species are characterized by a high content of renewable polymers.

In recent years, the research initiatives have resulted in a range of miscanthus traits being identified, which can be optimized for various applications. For example, improved miscanthus varieties for bio-based applications were released that are less recalcitrant to destruction due to having less lignin and due to alterations in specific cell wall characteristics [32]. In contrast, transgenic miscanthus with enhanced lignin content was derived in order to improve the energy value [33].

### 2.2. Studies on Environmental Impact of Miscanthus

Wang et al. [34] summarized publications in this field in their review paper. An economic model for the estimation of greenhouse gas emissions in the miscanthus cultivation using the commercial practice adopted in the UK was reported recently as well [35].

### 2.3. Production of Various Products from Miscanthus

A great many works worldwide have been focused on the miscanthus processing. Some applications employ all fractions of the miscanthus biomass, for example, incineration for power generation [36,37,38] or pyrolysis for the production of bio-oil [39,40], biochar [41,42], hydrochar [43,44] and graphene oxide [45], for biopolyol synthesis [27] and for the production of composite materials [46,47,48], concrete [49], miscanthus-based mortar [50], fiber-reinforced screed [51] and bio-based PET [52].

The other applications employ only certain parts of the cell wall for the transformation into products, for example, esterified lignin [53]. Acid hydrolysis of miscanthus has been studied for the synthesis of chemicals such as furfural, hydroxymethylfurfural [54], levulic acid [55] and other organic acids and ethylene glycol [56].

Cellulose, cellulose microfibers and paper [57,58,59,60], cellulose nanocrystals [61], oligosaccharides [62,63,64,65] and xylene [30] are derived from miscanthus. Pidlisnyuk et al. [66] comprehensively reviewed some products from miscanthus (agricultural products, insulation and composite materials, hemicelluloses, pulp and paper).

Many biotechnology products such as bioethanol, biogas, bacterial cellulose, enzymes, lactic acid, lipids, fumaric acid and polyhydroxyalkanoates are derived from miscanthus, as detailed in Section 3.

### 2.4. Miscanthus Pretreatment and Hydrolysis Processes

Furthermore, some studies are focused only on miscanthus pretreatment without end-product isolation [67]. The pretreatment of miscanthus biomass is highly requisite to obtain fermentable sugars and subsequent biotechnology products. Due to the heterogeneous structure, miscanthus has serious limitations with respect to the conversion and is recalcitrant to enzyme-assisted hydrolysis. The pretreatment step is chiefly meant to breakdown the structure composed of the three main renewable polymers, i.e., cellulose, hemicellulose and lignin, as well as minor non-structural constituents (extractives, ash).

Out of the three basic constituents, lignin is the most recalcitrant to degradation. Cellulose retains a significant crystallinity index and forms a rigid framework acting as a bearing structure of the cell wall. Hemicellulose, a heteropolymer of xylose, arabinose, galactose and other sugars, is not crystalline and therefore more amenable to hydrolysis than cellulose [68].

Similar to other lignocellulosic feedstocks, several pretreatment methods are applicable to miscanthus. Some methods are already reckoned to be conventional (ball milling, acid treatment, alkaline treatment, ammonia treatment, organosolv treatment, ionic liquid treatment, hot water treatment, steam explosion treatment), and new methods are under development (microwave, ultrasound, deep eutectic solvent, irradiation, high force-assisted pretreatment methods, biological pretreatment) [69,70]. That said, the conventional methods continue to be investigated for a deeper understanding of fractionation, optimization and process scale-up [71]. Furthermore, it is also proposed that a combination of two or more approaches for biomass pretreatment be used for maximum destruction of the biomass [72].

Figure 1 shows a schematic of the effect of pretreatment on biomasses [73].

The evaluation of different approaches demonstrates that successive efforts are still needed to develop an economical and eco-benign pretreatment strategy [68,72].

But, not all of the biotechnology products require that a biomass be pretreated; for instance, pretreatment is not mandatory for the biogas production and the use of lignocellulose as an inducer of enzyme production.

After pretreatment, cellulose and hemicelluloses can be hydrolyzed to monomeric sugars. Enzymatic hydrolysis of lignocellulosics is the most known and promising technique for biomass saccharification. Various hydrolytic enzymes produced by microorganisms are available in the market, as outlined in the tables below (columns “Enzymes for Hydrolysis”).

Enzymatic hydrolysis can liberate monomeric sugars in a very wide range, depending on the pretreatment method. For instance, Dai et al. [74] recently examined how pretreatment methods such as microwave, NaOH, CaO and microwave + NaOH/CaO influenced the sugar yield from miscanthus. The hexose yield showed a substantial range from 4.0 to 73.4% (% on a cellulose basis). The highest hexose yield was achieved by the 12% NaOH pretreatment and the lowest one by the 1% CaO + microwave pretreatment.

## 3. Biotechnology Products

### 3.1. Bioethanol

Nowadays, the role of bioethanol as a technical product in the world economy is constantly growing because bioethanol can be applied not only as an alternative eco-benign fuel or an additive, but also as a universal solvent and a precursor for the synthesis of a variety of chemicals. The biotechnological transformation of lignocellulosic feedstocks into bioethanol is fully compliant with the principles of circular economy and is in line with the concept of advanced development; therefore, the demand for bioethanol from that type of feedstock is growing sustainably [75].

Miscanthus holds the lead among different lignocellulosic feedstocks that are in use for second-generation bioethanol production.

Kang et al. [76] obtained bioethanol from *M. sacchariflorus* in a bench-scale plant. The ground miscanthus was pretreated with 0.4 M NaOH at 95 °C, with a biomass loading of 250 kg. The cellulose content of the resultant pulp increased from 37% to 51%, while the lignin content declined from 23% to 12%. Then, enzymes 30 FPU/g Cellic^®^ CTec2 and 15% Cellic^®^ HTec2 per the Cellic^®^ CTec2 amount added (Novo Inc., Bagsværd, Denmark) were used at the start of the reaction of the simultaneous saccharification and fermentation with pre-hydrolysis. The ethanol producer used was the commercial *Saccharomyces cerevisiae* yeast strain. A quite high bioethanol concentration of 45.5 g/L in the mash was achieved, but the ethanol yield per 1 ton miscanthus was as low as 165 L/t.

In the recent paper, Turner et al. [28] described bioethanol production from *M. × giganteus*. Miscanthus was pretreated with 1% H_2_SO_4_ and autoclaved at 121 °C. The subsequent enzymatic hydrolysis was performed at a solid loading of 12.5% using a freeze-dried Celluclast^®^ cellulase from *Trichoderma reesei* (Sigma-Aldrich). The hydrolyzate was fermented with *S. cerevisiae*. The result was 13.58 g/L ethanol from 35.13 g/L glucose, which is equivalent to 0.148 g/g dry mass of miscanthus.

We previously implemented a complete cycle of bioethanol production from *M. sacchariflorus* under pilot-industrial conditions [77]. The miscanthus pretreatment included mechanical comminution and treatment with 4% HNO_3_ in a 250 L vessel at 94−96 °C under atmospheric pressure. The pulp was obtained with a 37.4% yield and contained 90.0% hydrolyzables. The stage of simultaneous saccharification and fermentation with delayed inoculation was carried out in a 100 L fermenter at a solid loading of 100 g/L. An enzyme cocktail of CelloLux^®^-A (Sibbiopharm Ltd., Berdsk, Russia) and BrewZyme BGX (Tarchomin Pharmaceutical Works Polfa S. A., Warszawa, Poland) was utilized. The microbial producer used for bioethanol was *S. cerevisiae* yeast, with a nutrient broth consisting of salts and yeast extract being added to the fermentation medium. The result was the commercially promising ethanol concentration of ~40 g/L in the mash calculated as absolute ethanol, with the bioethanol yield being 260 L/ton miscanthus.

Hassan and Mutelet [26] reported a study on assessing how the pretreatment of M. × giganteus with deep eutectic solvents (DESs) would influence the bioethanol production. The DESs were prepared by intermixing choline chloride with glycerol, or ethylene glycol or urea. Miscanthus was pretreated with the DESs by adding 1 g Miscanthus to the mixture comprising 20% dimethyl sulfoxide (DMSO) and 80% DES. The cellulose-rich fraction obtained from the DES-pretreated miscanthus was subjected to the enzymatic hydrolysis and fermentation processes to yield bioethanol. The enzymatic hydrolysis employed β-glucosidase and cellulases from *T. reesei* (Celluclast 1.5 L, Sigma Aldrich). The fermentation of different solutions of the hydrolyzate was carried out with S. cerevisiae yeast. The best result was achieved when miscanthus was pretreated with mixed choline chloride/glycerol, and the fermentation medium in that experiment contained 18.03 g/L ethanol, which is equivalent to the yield of 138.4 g/kg miscanthus.

It is worth mentioning the known previous works (2017–2018) on bioethanol production from miscanthus, where the bioethanol yields attained 105–134 L/ton miscanthus [78] and 252–284 L/ton miscanthus [79].

Table 2 summarizes the results of miscanthus conversion into bioethanol in 2018–2022 and outlines some examples of bioethanol production from lignocellulosic feedstocks for a comparison purpose.

The above-listed examples of the bioethanol production differ greatly. Unfortunately, very few reports include such an important process parameter such as the bioethanol yield per feedstock unit. Consequently, a high bioethanol yield expressed in g/L does not always indicate the overall success of the process. The review papers [85,86] demonstrate that many species of regional feedstocks can successfully be converted into bioethanol, provided that high bioethanol yields can be achieved. The bast crops (hemp, kenaf, etc.) are less easily converted into bioethanol because their lignocellulosic matrix is particularly recalcitrant. In this respect, miscanthus is more preferred than the bast or woody crops, as it is more liable to an effective pretreatment

The striking example is the study by Zhang et al. [79], in which the ethanol production from five different herbaceous feedstocks was experimentally quantified for two annuals (corn stover and energy sorghum) and three perennials (switchgrass, miscanthus and mixed prairie). Even though the studied herbaceous feedstocks had very different characteristics, they showed a similar ethanol yield ranging from 200 to 327 L/t feedstock. That said, miscanthus holds the leading position in terms of the biomass yield per ha of an area under crop (14.36 t/ha), which is an important indicator. For this kind of feedstock, which is highly productive in itself, the authors suggest to focus on improving feedstock quality, “which can have a major impact on the biorefinery by increasing process ethanol yields and lowering the minimum ethanol selling price”.

### 3.2. Biogas

The partial replacement of natural gas by biogas is a critical goal under the climate change conditions. Anaerobic fermentation holds promise as a method for biogas production from renewable biomass resources, including miscanthus [87].

Thomas et al. [88] evaluated the methane production potential of eight *Miscanthus* genotypes: three genotypes of *M. × giganteus*, four genotypes of *M. sinensis* and one genotype of *M. sacchariflorus*. They also examined how alkaline pretreatments (NaOH and CaO) influenced the methane production from *M. × giganteus* Floridulus. The pretreatment conditions were as follows: 10 g alkali reagent per 100 g total solid, 200 g/L total solid loading, room temperature (23–26 °C), without mixing, with the duration and miscanthus particle size being varied. The untreated and pretreated miscanthus samples (solid and liquid fractions) were digested in batch anaerobic flasks. The flasks contained a sodium bicarbonate buffer, solutions of macro-elements and oligo-elements, an anaerobic sludge at 5 g volatile solids/L and substrate at 5 g total solid/L. The flasks were incubated at 35 °C for 60 days. As a result, the methane potential varied from 166 to 202 NmL CH_4_/g volatile solids among the eight *Miscanthus* genotypes studied. The genotype of the *M. sinensis* species showed the best productivity. The best methane productivity of +55% was achieved when miscanthus with a 1 mm particle size was pretreated with NaOH for 6 days. It was concluded that miscanthus selection could be directed towards deriving miscanthus intended for methane production. The alkaline pretreatment with high solid loading holds promise for improvements in methane production. Despite CaO having a lower performance than NaOH, lime should be considered due to the use of its digestate as a fertilizer and its better environmental impact.

Later, Jury et al. [89] confirmed that the lime pretreatment has a lower environmental impact. They evaluated the environmental impact of two alkaline pretreatments (lime and soda) of miscanthus and sorghum and their batch co-fermentation with manure. Miscanthus was shown to be able to lower climate change from −60 to −80%, but is inferior to sorghum (the climate change reduction varies from −90 to −105%).

Earlier studies described the methane production from miscanthus in the following yields (mL CH_4_/g volatile solids): from 229.5 (no pretreatment) to 327.4 (H_2_O_2_ pretreatment) [90] for *M. floridulus* without and with five pretreatment methods (NaOH pretreatment, H_2_O_2_ pretreatment, hot water pretreatment, microaerobic pretreatment, HCl pretreatment); 198.6 for *M. × giganteus* [91,92]; from 247.1 to 266.5 for *M. sinensis*, *M. floridulus*, *M. × giganteus* and *M. lutarioriparius* [92]; 190 for *M. sacchariflorus* and 100 for *M. × giganteus* [93]. These results are also outlined in the review paper by Song et al. [94] who highlighted the research into anaerobic fermentation of perennial energy grasses (pennisetum showed the best result and gave the methane concentration of up to 311 mL CH_4_/g volatile solids). As for lignocellulosic substrates from plant-based residues, their conversion into biogas is most widely studied because this process contributes to an efficient waste management and is the main source for bioenergy. Olatunji et al. [95] reviewed pretreatment methods for biogas generation from agricultural residues. A simple comparison between the concentrations of methane (mL CH_4_/g volatile solids) derived from those sources and miscanthus allows for the conclusion that miscanthus as a feedstock for biogas (without pretreatment) is superior to pinewood (38.7), rice straw (58.1), safflower straw (96.5%), corn stover (155.4) and reed biomass (188), and is almost comparable with wheat straw (210.4), sugarcane bagasse (222.2), barley straw (240) and meadow grass (297), and is significantly inferior to rapeseed stems and leaves (485.5) [95]. The outcomes of the anaerobic fermentation improve significantly if the feedstock is subjected to pretreatment, as also shown in Table 3. The pretreatment of miscanthus produced methane in a concentration up to 327.4 mL CH4/g volatile solids, exceeding most of the analogues. But the best result was achieved when wheat straw was subjected to the combined pretreatment of 0.7% NH_3_ and heat (538.1 mL CH_4_/g volatile solids) [95,96].

### 3.3. Bacterial Cellulose

Bacterial cellulose (BC) is a natural nanomaterial producible by some bacterial species. BC exhibits a high degree of crystallinity and a high degree of purity, as well as a unique structure consisting of the 3D network of ribbon-like nanofibers. The unique structure endows BC with properties such as a high tensile strength in the wet state, a large surface area, a high water-holding capacity and excellent permeability, flexibility, elasticity, durability, etc. BC is being used in many fields such as foods, personal care amenities, household chemicals, biomedicine, textile and composite resins. The economical and sustainable production of BC requires that the commercial sources of carbon (glucose) be replaced by renewable feedstocks [99].

Kim et al. [100] described BC production from miscanthus (alongside the other two lignocellulosic biomasses, barley straw and pine tree). The feedstocks were subjected to hydrothermal pretreatment in the presence of H_2_SO_4_ as the catalyst, and the resultant mixtures were detoxified by adsorption on activated carbon. The enzymatic hydrolysis of the pretreated biomass was run using Celic CTec2. The hydrolyzates were then diluted with distilled water to a 50 g/L glucose concentration, and supplements (corn extract, ammonium sulfate, dibasic potassium phosphate and magnesium sulfate heptahydrate) were added. The biosynthesis of BC was performed using *Gluconacetobacter xylinus* for 7 days. It is worth emphasizing that the miscanthus hydrolyzate exhibited the highest BC production in the test group (16.70 g/L, 97.86% of the control group). Most recently, this process was optimized [101], and the variation in some culture parameters resulted in BC with a bit lower concentration of 14.88 g/L but within a shorter time of 4 days.

We previously synthesized BC from the biomass of *M. sacchariflorus* growing in West Siberia [102]. The miscanthus biomass was subjected to one- and two-stage chemical pretreatments with 4% HNO_3_ and 4% NaOH under atmospheric pressure at 90−96 °C. The resultant four substrates were enzymatically hydrolyzed with CelloLux^®^-A and BrewZyme BGX, with a solid loading of 30 g/L. The enzymatic hydrolyzates were centrifuged, standardized against glucose to reach a glucose content of 20 g/L, heated to 100 °C, and used as extracting media to recover extractives from black tea. The biosynthesis of BC was run using symbiotic *Medusomyces gisevii* Sa-12 under static and non-sterile conditions for 24 days. The obtained yields of BC were relatively small, i.e., the best result was 1.24 g/L. It was speculated that this problem can be fixed by adapting the microbial producer to the enzymatic medium. In addition, the resultant BC was distinguished by a high index of crystallinity (88–93%) and an extraordinarily high content of allomorph Iα (99–100%), independent of the pretreatment method.

Our literature search has failed to find other examples of BC production from miscanthus, but there are reports describing the processes for BC production from other plant resources [103,104,105,106]. Table 4 summarizes the results of the miscanthus conversion into BC, as well as examples of BC production from other plant and wood raw materials among which miscanthus holds a leading position.

### 3.4. Enzymes

Enzymatic hydrolysis of cellulose is one of the bottlenecks for lignocellulosic biorefinery, and many studies are centered on the production of cellulase from *Trichodema reesei* [107]. Pure cellulose and sophorose, which are efficient inducers of *T. reesei*, are expensive substances. As an alternative, the sources from biomass composed of cellulose, hemicelluloses and lignin are being examined as low-cost inducers [108].

Xiang et al. [108] employed steam-exploded *M. lutarioriparius* as the inducer of cellulase production when fermented with *T. reesei*. The fermentation was run in a 20 L fermenter containing 9 L of medium and 1 L of culture. The fermentation medium comprised 40 g/L steam-exploded *M. lutarioriparius*, 1 g/L glucose, 2 g/L corn steep liquor, 2 g/L Tween-80 and salt supplements. Another 40 g/L steam-exploded *M. lutarioriparius* was fed into the fermenter with different feeding strategies at 48 h of fermentation using the solid screw feeder developed in that study. It was shown that cellulase production was more induced by *M. lutarioriparius* than by the other lignocellulosic biomass (corn or rape straw). The induction was considerably improved by continuous feeding compared to intermittent feeding. Consequently, the filter paper activity was 19.85 FPU/mL.

It should be noted that microcrystalline cellulose (22.73 FPU/mL) [109], coconut mesocarp (54 FPU/mL) [110] and mixed sugars derived from glucose by β-glucosidase-catalyzed transglycosylation (90.3 FPU/mL) [107] were able to provide a higher cellulase activity.

In the previous studies on miscanthus as the inducer of cellulase production [111,112], the resulting cellulase activity was below 2.0 FPU/mL [108].

The miscanthus biomass is also a good candidate for the manufacture of laccase that is considered to be a major enzyme to degrade lignin when the biomass is delignified. Guo et al. [70] cultivated the six efficient strains (AS1 (the genus of *Pseudomonas*), AS2B (the genus of *Exiguobacterium*), and A4, K1, X4 and X8 of the genus of *Bacillus*) to produce laccase on a mineral-salt medium from 0.5% *M. sacchariflorus* biomass as the single source of carbon. The peak laccase activity was detected on the first day in strains AS1 (8091 U/L) and K1 (1049 U/L). The maximum activity occurred on the third day in strains AS2B (2365 U/L) and X8 (103 U/L), on the fourth day in stain X4 (646 U/L) and on the fifth day in strain A4 (1122 U/L). For comparison, the maximum laccase activities induced by wheat bran in strains A4, AS2B, K1 and X4 were 246.7, 67.0, 82.4 and 137.0 U/L, respectively.

Table 5 summarizes all the latest results of enzyme production induced by miscanthus and outlines two examples of the other lignocellulosic feedstocks, the inducers of cellulase and laccase.

The inducible nature of enzyme production by various substrates is well known [113]. This explains the fact that a more reactive cellulase was achieved from the feedstock such as coconut mesocarp than from miscanthus. Since miscanthus contains quite a lot of lignin, it is understood that the activity of laccase derived from miscanthus is higher than that of laccase obtained from wheat bran.

The advances in producing enzyme complexes with a high enzymatic activity from miscanthus are of high application importance. It has been proved that the profitability of a bioethanol production plant improves greatly if the bioethanol production is combined with the production of enzyme complexes [114]. It can be said with confidence that this is also true for other bio-based products being produced from biomass, with a higher added value than bioethanol [115].

### 3.5. Lactic Acid

Lactic acid (LA) can be produced by chemical synthesis or microbial fermentation of sugars, and has a wide application in the cosmetic, food, pharmaceutical, medical and chemical industries. LA is a monomer of polylactic acid (a bioplastic), an essential product for the developing bioeconomy. Due to the high cost of refined sugars (simple sources of carbon for fermentation), there is currently a drive towards utilizing complex polymeric substrates as feedstock, such as lignocellulosic materials and food waste [116,117,118].

Recently, Gunes et al. [119,120] investigated LA production from the biomass of *M. × giganteus*. Miscanthus was pretreated with hot liquid water, and the solid fraction was then enzymatically hydrolyzed with the C2730 (synonyms: Celluclast^®^, Celluclast^®^ 1.5 L) cellulase enzyme from *T. reesei* (Sigma Aldrich). The hydrolyzate was used as a medium to produce LA by submerged fermentation with *Rhizopus oryzae* fungus. The maximum content of LA was found to be 6.8 g/L in 24 h, with a productivity of 0.28 g/L/h.

An interesting solution to LA production concurrent with bioethanol from *M. × giganteus* was previously suggested [120]. The miscanthus biomass was separated into liquid and solid phases by mechanical pressing. Bioethanol was prepared from the solid fraction after being pretreated with hot liquid water and dilute H_2_SO_4_ coupled with simultaneous saccharification and fermentation. The miscanthus juice was used to culture *Lactobacillus brevis* and *Lactobacillus plantarum*, and the co-fermentation produced a high LA concentration of 11.9 g/L.

As early as 2013, the DIREVO Industrial Biotechnology GmbH reported the miscanthus bioconversion by a pretreatment and a consolidated bioprocess using extremely thermophilic microorganisms of the genus *Thermoanaerobacter* [121,122].

Table 6 summarizes results of the miscanthus conversion into LA (2023) and outlines two examples of lactic acid (LA) production from other lignocellulosic raw materials. 

Judging from the LA production results from miscanthus, miscanthus is appreciably inferior to the other lignocellulosic feedstocks. In our view, it can be explained by the nature of miscanthus and by inhibitors of phenolic origin contained in it [125] because lactic-acid microorganisms are very demanding in terms of the composition of nutrient media. Unfortunately, there have recently been no studies on LA production, which were extensively performed in between 2006 and 2012. Even though new review papers emerge [116,117,118], they quote very few new references of LA production.

### 3.6. Lipids

Lipids, which can accumulate oleaginous microorganisms, are extractable and transformable into biodiesel. But, the manufacture of biodiesel is non-competitive compared to cheap fossil fuel. Even though it is believed that it is the lipid extraction stage that is the most expensive in the technology [126], the search for low-cost nutrient media for the cultivation of oleaginous microorganisms is also topical. Therefore, lignocellulosic biomass, including miscanthus, can be utilized as the source of sugars for the heterotrophic microbial production of lipids [127].

Martins et al. [127] used miscanthus as the carbon source to obtain intercellular lipids with *Rhodosporidium toruloides* yeast. Miscanthus was subjected to hydrothermal pretreatment (autohydrolysis). The pretreated miscanthus was then enzymatically hydrolyzed with Cellic^®^ CTec2. Further, *R. toruloides* were cultivated on undiluted and diluted miscanthus hydrolyzates in ratios of 1:4, 1:2 and 3:1. A nitrogen-limiting semi-defined culture medium was used for the dilution. The best result was achieved at a dilution ratio of 1:2 to furnish a biomass concentration of 6.3 g/L, a lipid content of 30.67% *w/w* of dry cell weight and a lipid concentration of 1.64 g/L. Those authors suggested that the described approach be used for the co-production from *R. toruloides* of lipids and carotenoids that are also of commercial interest.

Previously, Mast et al. (2014) [128] have already reported the use of non-detoxified hydrolyzates from miscanthus for lipid production using *Rhodotorula glutinis*, but the lipid production level was low (∼7% *w/w* of dry cell weight and 0.93 g/L lipid concentration).

Table 7 presents the latest result on the miscanthus conversion into lipids and outlines a few examples of lipid production from other plant raw materials.

The production of lipids from miscanthus is lower relative to the other plant feedstocks but is conceptually feasible and can be prospectively improved.

### 3.7. Fumaric Acid

Fumaric acid is a platform material for the synthesis of many chemicals (for instance, malic acid and L-aspartic acid), and can be polymerized to afford synthetic resins and biodegradable polymers. It is applied in the food and feed industries, and in the health and pharma industries. The conventional process for producing this acid involves catalytic isomerization of maleic anhydride which is a petroleum-based compound [131,132]. There is an alternative strategy for the bioproduction of fumaric acid by using fungi of the genus *Rhizopus*, and the carbon sources can be derived from lignocellulosic biomass [133].

The production of fumaric acid from glucose is a well-studied topic, whereas the option of using lignocellulosic hydrolyzates has received less attention. Swart et al. [134] have recently explored the use of a synthetic lignocellulosic hydrolyzate (mixed glucose and xylose) for the production of fumaric acid with *Rhizopus oryzae*. The continuous fermentation with a low mixed glucose/xylose feeding rate allowed fumaric acid to be produced in a 0.735 g/g yield. The glucose/xylose mixture is certainly far from authentic lignocellulosic hydrolyzates, but that study is a step towards a viable production of fumaric acid through the renewable, environmentally sustainable process.

Sebastian et al. [133] investigated the fumaric acid production from miscanthus using *R. oryzae*. Miscanthus was pretreated with 2% sodium hydroxide. The further enzymatic hydrolysis was run using Cellic-Ctec and Viscozyme-L (Sigma Aldrich). The production media for culturing *R. oryzae* consisted of a miscanthus hydrolyzate supplemented with salts and yeast extract. Consequently, the concentration and yield of fumaric acid produced with three *R. oryzae* strains were 8–9 g/L and 0.32–0.53 g/g reducing sugar (the theoretical yield of fumaric acid from glucose was 1.3 g/g). The study [133] also examined the fumaric acid production from switchgrass and hemp, “but miscanthus was identified as “the ideal perennial lignocellulosic biomass due to its higher reducing sugar concentration of 39.55 g/L” with a 79% enzymatic conversion of the biomass obtained after alkali pretreatment”.

To the best of our knowledge, the use of miscanthus hydrolyzates as feedstock for fumaric acid production has not been further reported.

### 3.8. Polyhydroxyalkanoates

Polyhydroxyalkanoates (PHAs) are green biopolyesters produced in nature by some microorganisms, and demonstrate a wide array of thermal, crystallization and mechanical properties. This versatility makes them attractive biomaterials suitable for applications and products that encompass consumer commodities, automobiles, healthcare, biomedicine, packaging, electronics, textile and 3D-printing materials [68]. The process for PHAs has a high prime cost, as it utilizes commercial sugars such as glucose, fructose and xylose.

Bhatia et al. [135] used a miscanthus biomass-derived hydrolyzate (along with a barley biomass hydrolyzate and a pine biomass hydrolyzate) as the source of carbon for the cultivation of *Ralstonia eutropha*. All the lignocellulosic biomass hydrolyzates prepared by dilute H_2_SO_4_ pretreatment and enzymatic digestion process were procured from Sugaren Co. Ltd. (Korea). *R. eutropha* was cultured in medium using hydrolyzates as substitutes for glucose under optimized conditions. Upon completion of cell growth, the biomass was separated from the supernatant for PHA extraction. The use of the miscanthus biomass-derived hydrolyzate resulted in a biomass accumulation of 4.6 g dry cell weight/L and a PHA production of 2.0 g/L. That said, the miscanthus biomass-derived hydrolyzate proved to be the best (1.8 g/L PHA from the barley biomass hydrolyzate and 1.7 g/L PHA from pine biomass hydrolyzate) [135].

It is exactly miscanthus for which no more examples of PHA production have been found; however, the topic of PHA production from other lignocellulosic sources has been under study for long, as described in the review paper by Govil et al. [68] and Sohn et al. [136]. If simply compared (i.e., conversion conditions are ignored), the concentrations of PHAs (g/L) derived from those sources and miscanthus allow for the conclusion that miscanthus is superior, for example, to wheat bran (0.319), pinus radiata wood (0.39) and rice straw (1.7), and is inferior to water hyacinth (4.3), bagasse (4.2), sunflower stalk (7.86) and oil palm empty fruit (12.48) [68].

Table 8 shows the result of the miscanthus conversion into PHAs and outlines a few examples of the PHA production from the other lignocelluloses.

## 4. Discussion

Among many miscanthus processing routes studied, the conversion into biotechnology products is a very popular trend. That said, the production of bioethanol and biogas has been well studied because these products are of great importance as alternatives to non-renewable fuels. The research on bacterial cellulose (BC) is growing in popularity because this polymer has unique properties and universal applications, most notably in medicine. The studies on the production of enzymes, lactic acid and lipids have a lower rating in popularity, while the works on the production of fumaric acid and polyhydroxyalkanoates (PHAs) from miscanthus are scarce.

Figure 2 illustrates a simplified formula that summarizes the production of biotechnology products from miscanthus and demonstrates the prospects of miscanthus as a universal feedstock for valuable products of different categories (biofuel, bio-based polymers, enzymes and platform molecules).

It is needless to say that the yields of biotechnology products are influenced by multiple factors. However, the comparison between the concentrations of the products from miscanthus and other lignocellulosic feedstocks, if the conversion conditions are ignored, gives some insights into the reasonability of the conversion. Miscanthus is more preferred for bioethanol, than the bast and woody crops, and is compatible with agricultural residues. Miscanthus as a feedstock for biogas is superior to resources such as pinewood, rice straw, safflower straw, corn stover and reed biomass, but wheat straw holds a leading position among the lignocelluloses for biogas. The concentration of bacterial cellulose obtained from miscanthus was higher than that from barley straw, pine tree, grape pomace and potato peel waste. With regard to the production of enzymes inducible by lignocellulosic feedstocks, coconut mesocarp was the inducer of a more active cellulase compared to miscanthus. But, miscanthus was the inducer of a more active laccase compared to wheat bran. miscanthus was identified as a promising lignocellulosic feedstock for the production of fumaric acid as compared to switchgrass and hemp. Miscanthus is superior in the PHA production to pinus wood, barley, wheat bran and rice straw, and is inferior to water hyacinth, bagasse, sunflower stalk and oil palm empty fruit.

The production of LA and lipids from miscanthus is lower than that from the other plant raw materials; but most importantly, the conceptual feasibility of producing those products from miscanthus is demonstrated.

It should be underscored that the present review was not aimed at comparing the yields of biotechnology products from miscanthus and other lignocellulosic resources. The biotechnological processes are sophisticated and multifactorial, and are influenced by many more factors in addition to a feedstock.

The advances made in the conversion of miscanthus into biotechnology products are of great significance. Even though most studies on the miscanthus conversion into biotechnology products are presently underdeveloped and miscanthus is inferior to some types of plant raw materials, its future holds promise. A further improvement in the fermentation processes using miscanthus as an alternative feedstock can completely change the structure of the industrial sectors of biofuel, biopolymers and other products, allowing the reduction of their production cost that can compete with petroleum-based and conventional resource-based fuel and polymers.

The issues facing the adjustment of efficient productions of goods from miscanthus necessitate the difficult choice of pretreatment methods and subsequent operations that rely on certain basic research. That said, the assessment of pretreatment methods and further process operations must include the material and energy balances and address the economic and environmental issues. This is required to efficiently evaluate the viability of processes. The life cycle assessment contributes to a more reliable evaluation of the sustainability of biorefinery systems, although there are some methodological problems as discussed in the review paper by Vance et al. [139].

The miscanthus conversion routes should also be examined for the simultaneous co-production of several products in view of enhancing the economic feasibility of the processes. The integrated bioconversion processes can maximize the profitability and sustainability.

It should be borne in mind that most of the techniques discussed in the literature were investigated under lab-scale conditions and cannot yield the same outcome when trialed on an industrial scale. Hence, it is necessary to examine these techniques on an industrial scale and devise better methods for the miscanthus conversion, starting with optimizing the pretreatment stage.

## 5. Conclusions

In this review, we provided an overview of the state-of-the-art research on the conversion of miscanthus macromolecules based on literature search results, primarily focusing on the last five years. This review represents the first comprehensive analysis of biotechnology products derived from miscanthus.

Given the number of works recently published over the last years, the biotechnology products can be arranged from the most popular to the least as follows: bioethanol, biogas, bacterial cellulose, enzymes (cellulase, laccases), lactic acid, lipids, fumaric acid and polyhydroxyalkanoates. The production of bioethanol and biogas from miscanthus is most studied, and miscanthus is among the best feedstocks for the listed products. The research on the other biotechnology products is underdeveloped, and some limitations associated basically with the nature of miscanthus and fermentation inhibitors contained in it need to be overcome.

The present review citing valuable works allows one to assess the miscanthus potential, comprehend deeper the conversion processes of miscanthus macromolecules and broaden the potentialities of these processes for the development of sustainable bioconversion technologies.

## Figures and Tables

**Figure 1 ijms-24-13001-f001:**
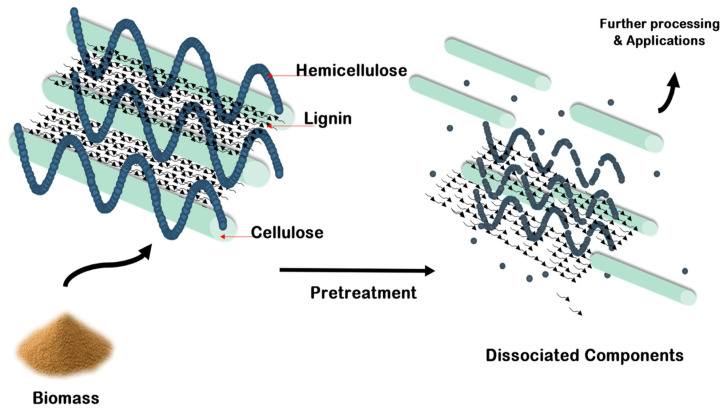
Effect of pretreatment on biomasses (reproduced with permission from [73], MDPI, 2023).

**Figure 2 ijms-24-13001-f002:**
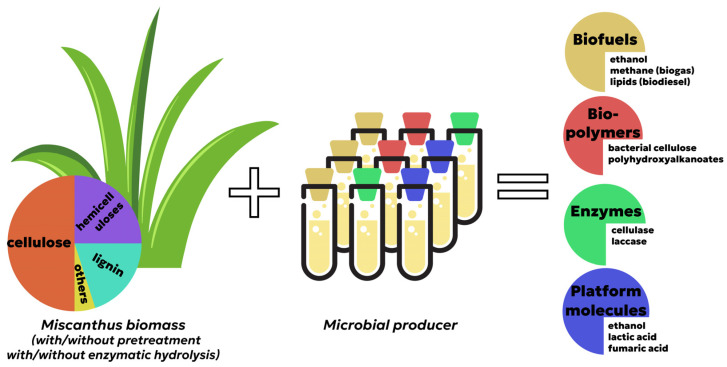
A simplified formula for production of biotechnology products from miscanthus.

**Table 1 ijms-24-13001-t001:** Component content (%) of miscanthus.

Miscanthus Species and Country of Habitat	Cellulose	Hemicelluloses	Lignin	Ash	Others	Ref.
*M. × gigantus* (France)	41.08	24.52	27.00	-	7.4	[26]
*M. × giganteus* (Canada)	55.01	17.42	16.90	-	10.67	[27]
*M. × giganteus* (UK)	45.5	29.2	23.8	-	-	[28]
*M. × giganteus* (different climate regions of Russia)	43.2–55.5	17.9–22.9	17.1–25.1	0.90–2.95	0.3–1.2%	[29]
*M. × giganteus* (Poland)	45.12	29.30	22.21	-	-	[22]
*M. sinensis* (Poland)	44.12	29.79	19.52	-	-	
*M. sacchariflorus* (Poland)	44.57	29.11	20.34	-	-	
*M. lutarioriparius* (China)	41.89	18.21	16.77	-	-	[30]
*M. sinensis* (Russia)	49.1	20.7	23.3	3.00	2.6	[31]
*M. sacchariflorus* (Russia)	53.3	21.3	28.1	5.66	2.4	
*M. sinensis* (China)	37.66 (av.), 48.52 (max.)	22.94 (av.)	17.35 (av.)	2.47 (av.), 4.5 (max.), 1.43 (min)	15.83 (av.)	[24]
*M. floridulus* (China)	36.28 (av.)	21.95 (av.)	16.94 (av.)	2.74 (av.)	18.41 (av.)	
*M. nudipes* (China)	36.07 (av.)	22.39 (av.)	17.21 (av.)	2.51 (av.)	21.10 (av.)	
*M. sacchariflorus* (China)	39.25 (av.),	26.35 (av.), 34.23 (max.)	18.11 (av.), 23.75 (max.)	2.51 (av.),	11.62 (av), 5.38 (min.)	
*M. lutarioriparius* (China)	39.96 (av.)	22.85 (av.)	18.69 (av.)	2.43 (av.)	12.43 (av.)	
Hybrid (China)	37.14 (av.)	21.21 (av.), 15.71 (min.)	16.37 (av.), 13.01 (min.)	2.56 (av.)	20.67 (av.), 34.88 (max.)	

**Table 2 ijms-24-13001-t002:** Results of the conversion of miscanthus and other lignocellulosic feedstocks into bioethanol.

Feedstock and Country of Habitat	Pretreatment	Enzymes for Hydrolysis	Microbial Producer	Ethanol Concentration and Yield	Year, Ref.
*M. sacchariflorus* (Korea)	0.4 M NaOH at 95 °C	Cellic^®^ CTec2 and HTec2	*Saccharomyces* *cerevisiae*	45.5 g/L, 165 L/t miscanthus	2019, [76]
*M. × giganteus* (UK)	1% H2SO_4_ and autoclaved at 121 °C	Celluclast^®^	*S. cerevisiae*	13.58 g/L, 0.148 g/g miscanthus	2021, [28]
*M. sacchariflorus* (Russia)	4% HNO_3_ at 94−96 °C	CelloLux^®^-A and BrewZyme BGX	*S. cerevisiae*	40 g/L, 260 L/t miscanthus	2022, [77]
*M. × giganteus* (France)	20% DMSO and 80% DES (Choline chloride/glycerol) at 373 K	Celluclast^®^ 1.5L	*S. cerevisiae*	18.03 g/L, 138.4 g/kg miscanthus	2022, [26]
*M. × giganteus* (USA)	AFEX-pretreatment at 100 °C	CTec2 and HTec2	*S. cerevisiae*	33.7 g/L, 252 L/t miscanthus	2018, [79]
			*Zymomonas mobilis*	38.0 g/L, 284 L/t miscanthus	
Corn stover (USA)			*S. cerevisiae*	32.1 g/L, 233 L/t corn stover	
			*Zymomonas mobilis*	45.1 g/L, 327 L/t corn stover	
Pine needle waste biomass (India)	1.0% NaOH + Microwave (900 W for 12 min)	Xylanase from *Bacillus pumilus* and cellulase from *Bacillus subtilis*	*Schizosaccharomyces* sp. EF-3 and *Kluyveromyces marxianus* (co-fermentation)	17.65 g/L	2022, [80]
Hemp (South Korea)	0.2–1.6% NaOH at 65 °C, 1% H_2_SO_4_ at 121 °C	Cellic^®^ CTec2	*S. cerevisiae*	18.9 g/L	2022, [81,82]
Kenaf (South Korea)				16.2 g/L	
Bamboo (*Phyllostachys edulis*) *(*China)	NaOH, acid catalyzed steam pretreatment 190 °C	Cellic^®^ CTec3 and β-glucosidase	*S. cerevisiae*	50.10 g/L	2020, [83,84]

**Table 3 ijms-24-13001-t003:** Results of the conversion of miscanthus and other lignocellulosic raw materials into biogas.

Feedstock and Country of Habitat	Pretreatment	Inoculum	Methane Concentration, mL CH_4_/g Volatile Solids	Year, Ref.
*M. sinensis* (France)	no	Anaerobic sludge (from UASB treating sugar industry wastewater)	202	2019, [88]
*M. sacchariflorus* (France)	no		195	
*M. × giganteus* Floridulus (France)	no		184	
	10% NaOH at 23−26 °C, 6 days		291	
	10% CaO at 23−26 °C, 6 days		245	
*M. floridulus* (China)	no	Biogas slurry, collected from biogas plant used corn straw as feedstock (China)	229.5	2018, [90]
	6% NaOH at 35 °C, 3 h		284.9	
	2% H_2_O_2_ at 35 °C, 24 h		327.4	
	Hot water at 95 °C, 10 h		260.0	
	Microaerobic pretreatment		271.6	
	HCl at 99 °C, 0.5 h		260.3	
Switchgrass (Turkey)	no	Anaerobic sludge	217.1	2022, [97]
	3% solid loading, 100 °C, 6 h		248.7	
Wheat straw (USA)	no	Inoculum from a mesophilic anaerobic digester in Pullman Wastewater Treatment Plant (USA)	407.8	2019, [96]
	0.7% NH_3_ and thermal at 105 °C		538.1	
Wheat straw (USA)	no	Effluent from a wastewater treatment plant (USA)	210.4	2018, [98]
	1% urea at 20 °C, 6 d		305.5	

**Table 4 ijms-24-13001-t004:** Results of the conversion of miscanthus and other plant and wood raw materials into bacterial cellulose (BC).

Feedstock and Country of Habitat (If Known)	Pretreatment	Enzymes for Hydrolysis	Microbial Producer	BC Concentration	Year, Ref.
Miscanthus (Korea)	hydrothermal pretreatment in the presence of H_2_SO_4_ + detoxified by adsorption on activated carbon	Celic^®^ CTec2	*Gluconacetobacter xylinus*	16.70 g/L	2021, [100]
*M. sacchariflorus* Maxim. (Russia)	two stages using 4% NaOH and 4% HNO_3_ at 90−96 °C	CelloLux^®^-A and BrewZyme BGX	*Medusomyces gisevii*	1.24 g/L	2021, [102]
Barley straw	hydrothermal pretreatment in the presence of H_2_SO_4_ + detoxified by adsorption on activated carbon	Celic^®^ CTec2	*G* *. xylinus*	13.09 g/L	2021, [100]
Pine tree				12.54 g/L	
Grape pomace + potatoes (Spain)	2% H_2_SO_4_ at 125 °C + neutralization with CaCO_3_	no	*Komagateibacter xylinus*	4.0 g/L	2022, [105]
Potato peel waste	2.0 M of each of nitric, sulfuric, hydrochloric and phosphoric acid at 100 °C for 2, 3, 4 and 6 h	no	*G* *. xylinum*	4.7 g/L	2019, [106]

**Table 5 ijms-24-13001-t005:** Results of enzyme production induced by miscanthus and other lignocellulosic feedstock types.

Feedstock and Country of Habitat (If Known)	Pretreatment	Microbial Producer	Biotech Product	Enzyme Activity	Year, Ref.
*M. lutarioriparius* (China)	steam explosion at 195 °C, 10 min	*T. reesei*	Cellulase	Cellulase activity 19.85 FPU/mL	2021, [108]
*M. sacchariflorus*	no	*Pseudomonas*	Laccase	Laccase activity 8091 U/L	2019, [70]
Coconut mesocarp (India)	liquid hot water treatment at 210 °C for 20 min + 0.5% NaOH at 180 °C for 40 min	*T. reesei*	Cellulase	Cellulase activity 54 FPU/mL	2018, [110]
Wheat bran	no	*Bacillus* sp.	Laccase	Laccase activity 246.7 U/L	2019, [70]

**Table 6 ijms-24-13001-t006:** Results of the conversion of miscanthus and other lignocellulosic feedstocks into lactic acid (LA).

Feedstock and Country of Habitat	Pretreatment	Enzymes for Hydrolysis	Microbial Producer	Lactic Acid Concentration	Year, Ref.
*M.× giganteus* (Turkey)	LHW at 140 °C, 100 bar and 45 min	C2730	*Rhizopus oryzae*	6.8 g/L	2023, [119]
Sugarcane bagasse (Australia)	0.72% H_2_SO_4_ 170 °C for 15 min + steam explosion	Genencor GC220 (Denmark)	*Bacillus coagulans*	70.4 g/L	2016, [123]
Corn stover (China)	simultaneous bio-delignification and saccharification with lignocellulolytic enzyme system obtained from co-fungi culture		*B* *. coagulans*	92 g/L	2016, [124]

**Table 7 ijms-24-13001-t007:** Results of the conversion of miscanthus and other feedstocks into lipids.

Feedstock and Country of Habitat (If Known)	Pretreatment	Enzymes for Hydrolysis	Microbial Producer	Lipids Concentration and Lipid Content	Year, Ref.
Miscanthus (Netherlands)	hydrothermal pretreatment at 190 °C for 15 min	Cellic^®^ CTec2	*Rhodosporidium toruloides*	1.64 g/L, 30.67% of cell dry weight	2021, [127]
Cardoon stalks (Italy)	0.6% H_2_SO_4_ solution for 10 min + steam explosion at 195 °C, 7.5 min	CTEC2	*Solicoccozyma terricola*	13.20 g/L, 55.60% of cell dry weight	2018, [129]
Residues from olive tree pruning (Italy)	Steam explosion at 210 °C for 25 min	NS-22192 (Novozyme, Denmark)	*Naganishia adeliensis*	4.90 g/L, 44.38% of cell dry weight	
Corn stover	AFEX pretreatment at 140 °C for 30 min	Cellic^®^ Ctec3 and Cellic^®^ Htec3	*Cryptococcus humicola*	15.5 g/L, 40% of cell dry weight	2014, [130]

**Table 8 ijms-24-13001-t008:** Results of the conversion of miscanthus and other lignocellulosic feedstock into polyhydroxyalkanoates (PHAs).

Feedstock	Pretreatment	Enzymes for Hydrolysis	Microbial Producer	PHAs Concentration	Year, Ref.
Miscanthus	Hydrolyzate prepared by dilute H_2_SO_4_ pretreatment and enzymatic digestion		*Ralstonia eutropha*	2.0 g/L	2019, [135]
Barley				1.8 g/L	
Pine				1.7 g/L	
Sunflower stalk	Hydrothermal treatment at 190 °C for 5 min	Cellic^®^ CTec3	*R*. *eutropha*	7.86 g/L	2016, [137]
Wheat bran	1% NaOH	Commercial cellulase of *T. reesei* and β—glucosidase of *Aspergillus niger*	*R*. *eutropha*	0.319 g/L	2016, [138]

## Data Availability

Not applicable.

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
