# Peer review of "Recent Advances in Miscanthus Macromolecule Conversion: A Brief Overview"

_ijms, 2023, doi:10.3390/ijms241613001_

Round 1

Reviewer 1 Report

The authors have considered in detail possibilities and expediency of production of several products from Miscunthus biomass. The key products are cellulose, hemicellulose and lignin, which can be converted by enzymatic hydrolysis into biotechnology products. Among them, such useful and popular products as biofuel and biogas (ethanol, methane, some lipids for diesel) are the most important. In addition, very promised approaches to synthesis of bacterial cellulose, lactic acid, fumaric acid, polyhydroxyalkanoates, some enzymes (cellulase, laccase) are also considered. There are several kinds of Miscantus plants and authors very attentively discussed advantages and disadvantages every kinds of them for various applications. The review is constructed in such a way, when discussion of preparation from Miscanthus one of the other products are summarized in two figures. The first one is dedicated to “effect of pretreatment on biomasses” as the introduction to the problem. The second one concerns the final version, i.e. conclusion of the extraction from 97 references that quite enough for the review.      Unfortunately, the simple enumeration of different sources gives to readers some schematic image of the scientific activity in the field of Miscanthus processing. I believe that authors should accent attention on comparison not only biomasses from wood and plant sources, but underline that Miscanthus has advantages compared with other plants, such as hemp and flax, because it is multi-year, but others are annual. Besides, the composition of Miscanthus in particular cellulose content also has advantage compared with other plants.

And the last note (or advice): cellulose from Miscanthus is suitable for processing to textile fibers and authors should mention this feature of this plant.     

The English is understandable, but needs some polishing by language owners. 

Author Response

Please find atatched.

Reviewer 2 Report

The detail comments are attached.

English is good. 

Reviewer 3 Report

English is fine. Of course, it is deteted as a style from non native Scientific English.

Since it is a review, it is worthy if EACH subsection summarizes and compares the data with other sazonal and wood materials processing , say "2.1. Miscanthus Selection" deserves a table with the material physical and chemical characteristics including the diversity along the geographycal location, "2.3. Production of Various Products from Miscanthus" needs a table summarizing all type of products/processing, brief coments,  with aditional column quoting references. After "3.2. Biogas" a Table with main processing characteristics and main results from literature and the  comparison with another raw materials, and so on, and so on.
